# Optimal microRNA Sequencing Depth to Predict Cancer Patient Survival with Random Forest and Cox Models

**DOI:** 10.3390/genes13122275

**Published:** 2022-12-02

**Authors:** Rémy Jardillier, Dzenis Koca, Florent Chatelain, Laurent Guyon

**Affiliations:** 1Univ. Grenoble Alpes, CEA, Inserm, IRIG, BioSanté U1292, BCI, 38000 Grenoble, France; 2Univ. Grenoble Alpes, CNRS, Grenoble INP, GIPSA-Lab, Institute of Engineering University Grenoble Alpes, 38000 Grenoble, France

**Keywords:** sequencing depth, cancer, microRNA, survival, Cox model, random survival forest model

## Abstract

(1) Background: tumor profiling enables patient survival prediction. The two essential parameters to be calibrated when designing a study based on tumor profiles from a cohort are the sequencing depth of RNA-seq technology and the number of patients. This calibration is carried out under cost constraints, and a compromise has to be found. In the context of survival data, the goal of this work is to benchmark the impact of the number of patients and of the sequencing depth of miRNA-seq and mRNA-seq on the predictive capabilities for both the Cox model with elastic net penalty and random survival forest. (2) Results: we first show that the Cox model and random survival forest provide comparable prediction capabilities, with significant differences for some cancers. Second, we demonstrate that miRNA and/or mRNA data improve prediction over clinical data alone. mRNA-seq data leads to slightly better prediction than miRNA-seq, with the notable exception of lung adenocarcinoma for which the tumor miRNA profile shows higher predictive power. Third, we demonstrate that the sequencing depth of RNA-seq data can be reduced for most of the investigated cancers without degrading the prediction abilities, allowing the creation of independent validation sets at a lower cost. Finally, we show that the number of patients in the training dataset can be reduced for the Cox model and random survival forest, allowing the use of different models on different patient subgroups.

## 1. Introduction

microRNAs (miRNAs) are near 22-nucleotide long RNAs repressing protein-coding gene expression at post-transcriptional level [1]. miRNAs have been shown to be implicated in various steps of carcinogenesis: initiation, propagation and metastasis [2]. The cancer genome atlas (TCGA) project has provided microRNA sequencing on thousands of tumor samples over 33 cancer types, together with patient follow-up [3]. miRNAs represent promising biomarkers to predict patient survival in cancer [4]. TCGA datasets are extremely valuable to build survival models with tumor miRNA expression, and contain large enough cohorts for many tumor types to evaluate the predictions. A semi-parametric and popular model to link patient survival with genomics variables, dealing with censored data and assuming proportional hazards, has been proposed by D.R. Cox [5]. Classically, in the case of high-dimensional datasets, a penalty term is used to constrain the coefficients of the model, and to select only a subset of genes. Different forms of penalties exist [6], but we will focus on the elastic net penalty [7] in this paper, as we have recently shown that they provide similar performances [8]. More recently, non-parametric machine learning algorithms have been proposed and adapted to deal with survival data, including random survival forest [9]. Random survival forest potentially offers more flexibility, as it does not assume any proportionality between hazards, and takes into account non-linear effects and interactions between variables [10,11].

Building and evaluating efficient models from miRNA expression, applicable in clinics, requires to build large datasets from patient cohorts. It implies the recruitment of many patients, and the tumor miRNA profiling at a high enough sequencing depth. Increasing the number of patients and/or the sequencing depth means increasing the cost, but may not lead to direct improvement of the prediction performance of the models. Moreover, validation datasets remain scarce and expensive to build [12]. Thus, a compromise has to be found. For tumor mRNA profiling, P. Milanez-Almeida et al. [13] showed that the sequencing depth could be decreased by typically two orders of magnitude for TCGA datasets for most cancers when using the Cox model with elastic net penalty. They use C-index and *p*-value from single variable Cox model as prediction performance metrics. They argue that the saved cost could be used to increase the number of patients and/or to perform longitudinal studies.

The goal of the present work is to investigate the required miRNA and mRNA sequencing depth together with the number of patients in the training dataset to build optimal performance models according to well-established metrics (i.e., C-index and integrated Brier score), both with the classical Cox model with elastic net penalty and random survival forest. Additionally, we have validated our results with an independent cohort.

## 2. Materials and Methods

### 2.1. Overview of the Methodology

Appendix A shows the methods as a flowchart. Briefly, sequencing data from TCGA are down-sampled (see Section 2.6.1), and then a cross-validation procedure is performed (as detailed in Section 2.3). Learning of the model is performed on 80% of the patients, eventually further reduced (see Section 2.6.2), and testing the performance on the remaining 20% is performed using the C-index and the Integrated Brier Score (IBS). Subsampling is performed both on the training set and on the testing set for TCGA datasets, which aims to answer the question “can we reduce both the sequencing depth for learning models and for patients in hospital?”.

### 2.2. Cox Model with Elastic Net Penalty and Random Survival Forest: The Link between Genetic and Survival Data

#### 2.2.1. Cox Proportional Hazards Model with Elastic Net Penalty

Let *T* denote the survival time (also called the ‘time-to-event’). The Cox model [5] is widely used in medicine to link covariates to survival data through the hazard function, defined for all time instants t>0, $h\left(t\right)=\lim_{h\to 0}\frac{P(t\leq T<t+h|T\geq t)}{h}$, which represents the instantaneous death probability per unit of time. In the Cox model, the hazard function for patient *i* is modeled as follows:$$h\left(t;X^{i}\right)=h_{0}\left(t\right)\exp\left(\beta^{T}X^{i}\right),$$
where $h_{0}\left(t\right)$ is the baseline hazard function, $X^{i}={\left(X_{1}^{i},\ldots ,X_{p}^{i}\right)}^{T}$ the vector of covariates for patient *i* (here as mRNA or miRNA expression, with *p* the number of coding or miRNA genes), and $\beta ={\left(\beta_{1},\ldots ,\beta_{p}\right)}^{T}$ the vector of associated coefficients. We define $\Delta_{i}$ to be the associated status, as 1 for death and 0 for censoring. The vector of coefficients β can be estimated by maximizing the Cox pseudo-likelihood, as proposed by Breslow [14].

The elastic net methodology [7] consists of the addition of a penalty term to the log-pseudo-likelihood l(β) before the maximization:$$\hat{\beta }\left(EN\right)=\underset{\beta }{\mathrm{argmax}\;}l\left(\beta \right)-\lambda \left({\alpha ||\beta ||}_{1}+\frac{1-\alpha }{2}{\left||\beta |\right|}_{2}^{2}\right)$$

In a previous work, we advised equally Lasso, Elastic Net and Ridge as penalization methods, as they provide equal model performance [8]. The Lasso selects fewer genes, leading to a more parsimonious model. In the present work, we chose the Elastic Net penalty as it is becoming the most popular penalization for Cox models.

We used the R package *glmnet* [15] to estimate Cox model with elastic net penalty. In the followig, ‘Cox model’ refers to ‘Cox model with elastic net penalty’. For more details on the Cox model and the choice of the hyperparameters used for penalties (i.e., α=0.3), we refer the reader to Appendix A.

#### 2.2.2. Random Survival Forest

Random forest, introduced by Breiman, is a classical ensemble algorithm for regression and classification whose principle is to build multiple decision trees and create a forest [16]. Results are averaged over all the trees. H. Ishwaran et al. then extended the classical random forest algorithm to survival analysis with censored data [10]. At each node of each tree, *m* explanatory variables are randomly chosen, and the variable that best separates patients into two groups according to their survival curve is retained. Tree depth is controlled by a threshold on the minimum number of patients in the node. Random survival forest has the advantage of possibly taking into account non-linear effects and interactions between variables [10,11].

We used the R package *tuneRanger* [9] to learn random survival forest for survival data, a package based on the *ranger* package [17] but with a fast implementation for tuning the number *m* of variables randomly drawn at each node. We used default hyperparameters suggested by the authors (i.e., patients used to build a tree chosen with boostrapping, at least 3 patients in a terminal node, 50 trees in a forest, log-rank test as splitting rule, $\sqrt{p}$ as starting value for tuning *m*, with *p* the total number of genes), and the function *tuneMtryFast*. To decrease computation time for mRNA datasets, we only retain the 2500 genes with the highest association with survival according to likelihood tests in single-variable Cox models (i.e., we learn one single-variable Cox model for each gene to compute the *p*-values).

### 2.3. Prediction Performance Metrics

As schemed in Appendix A, we estimate the prediction performance of the models by 10 repetitions of K-fold cross-validation (K=5). We learn a model (i.e., Cox model or random survival forest) on a training dataset ($\frac{4}{5}$ of the patients), and we define a risk score from this estimation for each patient of the testing dataset ($\frac{1}{5}$ of the patients). The risk score (RS) is defined for a given patient *i* as the sum of $\beta_{j}X_{j}^{i}$ for the Cox model ($X_{j}$ corresponds to the expression of gene *j*), and the mean of the estimated cumulative hazard function for the random survival forest:${\hat{\mathrm{RS}}}_{i}={\hat{\beta }}^{T}X^{i}$ for the Cox model, with $\hat{\beta }$ the estimator of the coefficients, and $X^{i}$ the gene expression vector for patient *i*.${\hat{\mathrm{RS}}}_{i}=\frac{1}{\mathrm{Card}(T)}\sum_{j\in T}\hat{H}\left(t_{j}|X_{i}\right)$ for random survival forest, with ‘Card’ the cardinal function, $\hat{H}\left(t|X_{i}\right)$ the estimated cumulative hazard function at time *t* for patient *i*, and T the times at which the hazard function is estimated.

This procedure allows to assess prediction performance by computing the C-index and the Integrated Brier Score (IBS), as defined below. Then, at the end of the procedure, 50 C-indices and 50 IBS are computed for each method.

The C-index allows the discrimination ability of a model to be assessed by quantifying the proportion of patient pairs for whom risk scores are in good agreement with their survival data. For two patients *i* and *k* with risk scores ${\mathrm{RS}}_{i}$ and ${\mathrm{RS}}_{k}$, and with survival times $T_{i}$ and $T_{k}$, the C-index is defined as $C=P(T_{i}<T_{k}\;|\;{\mathrm{RS}}_{i}>{\mathrm{RS}}_{k})$. A C-index of 1 indicates perfect agreement, and a C-index of $\frac{1}{2}$ corresponds to random chance agreement. We took the estimator of the C-index given by [18] and theorized by [19].

The Brier Score [20] measures the average squared distance between the observed survival status and the predicted survival probability at a particular time *t*. It is always a number between 0 and 1, with 0 being the best possible value. We used the IBS that integrates the Brier Score between 0 and the maximum event time of the test set, and divides this quantity by the maximum integration time. Then, while the C-index measures the ability of a model to rank patients according to their risks, the IBS estimates the ability of a model to predict survival probabilities along time. The IBS is a global performance metric that assesses both discrimination and calibration. These two metrics are widely used to estimate prediction performance in practice and are complementary.

We used the R packages *survcomp* [21] to compute the C-index, and *pec* for the IBS [22].

### 2.4. The Cancer Genome Atlas and E-MTAB-1980 Datasets

Cancer acronyms, as provided by the TCGA consortium, are available in Appendix A. First, we included cancers available in TCGA for which there were more than 75 patients with miRNA-seq and survival data. Then, we followed recent formal recommendations [23] to exclude the PCPG cancer that has too few death events and the SKCM cancer that has a high ratio of metastatic samples sequenced. We used overall survival as the disease-outcome, except when the authors recommend the use of progression-free interval (BRCA, LGG, PRAD, READ, TGCT, THCA and THYM). After these two steps, we retained 25 cancers. Finally, we computed the C-index and IBS estimates after running the Cox model or random survival forest applied on the miRNA profiles for these 25 cancers as schemed in Appendix A. To focus on cancers for which the sequencing data convey prognostic values, we decided to retain only the datasets for which the median C-index is significantly higher than 0.6 for at least one of the algorithms (i.e., Cox model or random survival forest) according to a one-sided Wilcoxon test at level 0.05. At the end of this procedure, we retained 11 cancers (Table 1, Appendix A). Appendix A shows the median miRNA sequencing depth and the number of patients for the 25 cancers. There is no statistical difference between the retained 11 cancers and the remaining 14 cancers, both in sequencing depth (*p* = 0.17, Wilcoxon test) and the number of patients (*p* = 0.85).

We used the Broad GDAC FIREHOSE utility (https://gdac.broadinstitute.org/ (accessed on 11 December 2020)) to obtain clinical, miRNA-seq, and mRNA-seq datasets. We applied a Trimmed Mean of M-values (TMM) procedure to correct for between sample variance [24]. We first used the *calcNormFactors* function of package *EdgeR* [25] to compute a normalization factor for each patient, and we then applied the *voom* function of the *limma* package to compute log2-CPM data corrected with the normalization factors computed earlier [26]. We then standardized the expression of each gene both in the training dataset and the testing datasets using the mean and standard deviation values among patients of the training data.

To confirm the impact of sub-sampling on survival metrics in an external validation cohort, we acquired the processed E-MTAB-1980 dataset [27] from ArrayExpress (https://www.ebi.ac.uk/arrayexpress/experiments/E-MTAB-1980/ (accessed on 25 November 2019)). Survival metrics were calculated as previously described, while using E-MTAB-1980 as a testing dataset. We standardized the testing dataset independently from the training dataset, by using the mean and standard deviation of the testing dataset. This dataset gathers transcriptomic and clinical data from 101 patients of a Japanese cohort. The transcriptomic data were acquired with a microarray technology, which is a continuous technique, so we have not applied any down-sampling. The goals with this independent dataset are: “can the approach with cross-validation on the TCGA data be validated with an independent dataset?”, and “to which extent down-sampling affects the model quality” (keeping unchanged the testing set).

### 2.5. Integration of miRNA-seq Data Together with Clinical Data

We verified whether the tumor miRNA profiles added predictive value to the clinical data [28]. Different strategies exist for integration of miRNA-seq and clinical data [29,30]. In order to avoid the dilution of the few clinical parameters among all the miRNA covariates, we added the risk scores computed with miRNA-seq data alone (${\mathrm{RS}}_{miRNA}$) to ones computed using classical clinical features (age, gender, grade, T, N, M), when available ($\mathrm{RS}=\beta .{\mathrm{RS}}_{miRNA}+\sum_{l}\beta_{l}.{\mathrm{Clin}}_{l}$, where ${\mathrm{Clin}}_{l}$ is the $l^{th}$ clinical variable). T is a score which stands for the extent of the tumor, N for the extent of spread to the lymph nodes, and M for the presence of metastasis. We did not include gender for sex-specific cancers (CESC, UCEC, PRAD). Age is available for all cancers, and we specify whether the other variables are available in Appendix A.

To emphasize if the miRNA-seq data added prediction value over clinical data for both Cox model and random survival forest, we performed a one-sided Wilcoxon signed rank test for each of the 11 cancers studied. We considered a difference significant when the *p*-value corrected with Benjamini-Hochberg method is below 0.05, even though this is purely indicative as discussed below.

### 2.6. Degradation of miRNA-seq Data

#### 2.6.1. Subsampling of miRNA-seq Data

“Sequencing depth” is defined here as the sum of the number of aligned reads per patient, and can vary according to the patients, and is equivalent to the notion of “library size”. These two nomenclatures will be used interchangeably in the following text.

To reduce the sequencing depth, we used a subsampling method [31]. The key parameter to calibrate fold reduction is the proportion of subsampling, ε∈(0,1]. For each count data (i.e., number of reads) $R_{ij}$ obtained for a patient *i* and a gene *j*, a subsampled count data of a proportion ε, noted ${\tilde{R}}_{ij}$, is drawn according to a binomial distribution of parameters $R_{ij}$ and ε:$${\tilde{R}}_{ij}∼\mathrm{Binom}\left(R_{ij},\varepsilon \right),$$
for each patient i=1,…,n and each gene j=1,…,p.

Thus, the closer the parameter ε is to 0, the smaller the read depth: a proportion of ε (e.g., 0.01) corresponds to a subsampling of the sequencing data by a factor $\delta =\frac{1}{\varepsilon }$ (e.g., 100). In this study, we examine the effect of 1 (no subsampling), 10, 100, 1000 and 10,000 subsampling factors δ. We then draw saturation curves, which are the evaluation metrics (i.e., C-index, IBS), performed on the test set which is not subsampled, as a function of the subsampling factor δ [32,33].

#### 2.6.2. Reduction of the Number of Patients in the Training Dataset

To study the impact of the number of patients on prediction capabilities, we artificially decreased the percentage *x* of patients in the learning dataset. In this study, we chose x=10,20,…,80%. However, to ensure that the C-index and IBS are not biased, the testing dataset is always composed of 20% of patients.

## 3. Results

### 3.1. Library Sizes of mRNA-seq Data Are Ten Times Larger Than the Ones of miRNA-seq Data

The library sizes are equivalent between the 25 cancers, with a few exceptions, and are distributed around $5\times 10^{6}$ reads for miRNAs, and $5\times 10^{7}$ for mRNAs (Appendix A). Recall here that only aligned reads, and not raw reads, are taken into account. The sequencing depth for mRNA datasets is therefore higher than that of the miRNAs by a factor of 10 on average, which is not surprising as it spans on 40 times more genes. There are thus, on average, 4 times more aligned reads per gene for miRNAs than for mRNAs. The lengths of genes are also very different between mRNAs and miRNAs. Also, there is no particular relationship between the sequencing depth chosen for the mRNAs and for the miRNAs between the different cancers. Note that for the LAML cancer, we observe a lower sequencing depth than for the other cancers: the median sequencing depth is 720,000 reads for LAML, 2.5 million for KIRC and 7.5 million for LGG (Appendix A).

### 3.2. C-Index Highlighted Noticeable Prediction Differences between Cox and Random Survival Forest Models for Eight out of Twenty-Five Cancers

Using miRNA-seq datasets and according to the C-index metric, the Cox model shows better prediction than the random survival forest for KIRC, CESC, PRAD, LUAD, HNSC, and to a lesser extent LIHC (Appendix A). Conversely, random survival forest shows better predictions for KIRP, THYM, THCA, and to a lesser extent UCEC. For the other cancers, we did not observe clear differences. Noticeably, while the Cox model is not able to capture any prediction abilities for THCA (i.e., median C-index of 0.46), random survival forest exhibits a median C-index of 0.62. However, if we choose the IBS as the prediction metric, random survival forest shows better prediction than Cox except for LGG (Appendix A). We discuss this difference observed between the metrics below.

### 3.3. mRNA-seq Data Provides Slightly Better Prediction Performance Than miRNA-seq Data for Most of the 11 Investigated Cancers

In this section, we use a Wilcoxon signed-rank test to highlight the situation in which there exist differences, and we corrected the 11 *p*-values computed with the Benjamini-Hochberg procedure.

The median C-indices reached with the Cox model are higher with mRNA-seq data than miRNA-seq data for all selected cancers except LUAD. More precisely, the C-indices are higher with mRNA-seq data for 8 cancers (ACC, KIRP, MESO, KIRC, LGG, CESC, PRAD, UCEC), and higher with miRNA-seq data only for LUAD (Appendix A). When using IBS as a metric, the median IBS obtained from 5 cancers (UVM, KIRC, LIHC, LUAD, UCEC) was lower compared to the median IBS obtained with mRNA-seq data. Additionally, overall IBS obtained by using mRNA-seq data was lower in cases of KIRP, MESO, LGG, and CESC, and higher while using miRNA-seq data of UVM, KIRC and LIHC. Comparable results were obtained using random survival forest as prediction model. Overall, mRNA-seq data provides better predictions, but the absolute differences remain small.

### 3.4. Mirna-seq Data Improves Predictions over Clinical Data Alone for Most of the Investigated Cancers

For 8 cancers (UVM, ACC, MESO, LGG, CESC, LIHC, PRAD, LUAD) out of the 11 studied, the addition of miRNA-seq data to generic clinical data significantly improved the C-index compared to clinical data alone for the Cox model (Appendix A, integration of clinical and miRNA data is described Section 2.5). Similarly, for random survival forest, the median C-index is improved for 6 cancers (UVM, ACC, MESO, LGG, LIHC and PRAD, Appendix A). When using the IBS as the performance metric, the difference are often not as clear: predictions appear better for 5 cancers when taking tumor miRNA profiles into account in the Cox model (KIRP, MESO, KIRC, LGG, and CESC, Appendix A), and for 5 cancers in the random survival forest model (ACC, KIRP, MESO, LGG, and LIHC, Appendix A).

Overall, the addition of miRNA-seq data to classical clinical data improves prediction performance as assessed by C-index and/or IBS for all the 11 cancers investigated but UCEC with the Cox model. This performance drops to 7 cancers with random survival forest; KIRC, CESC, LUAD, UCEC do no show improvement. The use of miRNA-seq data seems not as interesting as clinical data alone to build predictive risk scores for UCEC. However, we have included this cancer because RNA-seq data can be used in other contexts as part of a survival model: stratifying patients according to transcriptomic profiles [34], identifying predictive markers of response to treatments [35], identifying potential therapeutic targets [36].

### 3.5. Shallow Tumor miRNA or mRNA Sequencing Keeps Survival Prediction Performance for Many Cancers

We consider that the sequencing depth can be reduced if and only if none of the two prediction metrics (i.e., C-index or IBS) is degraded at level 0.05 according to a one-sided Wilcoxon test. Figure 1 shows the C-index as a function of miRNA (or mRNA) library size reduction and/or number of patients subsampling for the kidney cancer subtype KIRC (corresponding to clear cell renal cell carcinoma, ccRCC). We highlight this cancer subtype for the following reasons: the dataset contains many patient data (Table 1), the prediction performance is quite high (C = 0.7), the availability of an independent dataset (Section 3.8), and as we are more specialized on this subtype [37]. For this cancer, it appears that both the number of patients and the sequencing depth could have been decreased while keeping similar prediction capacity. More precisely, using 60% of the patients (n=304) in the training set leads to similar model performance, even though a small but noticeable performance decrease is noticed with IBS in the Cox model (Appendix A). Also, decreasing the miRNA and mRNA sequencing depth by one order of magnitude has no measurable consequences. However, both the number of patients and the sequencing depth should not be decreased to their maximum extents altogether, as shown by the shape of the color map (Figure 1A).

Table 2 and Table 3 summarize the maximum lowering of sequencing depth without affecting prediction performance. For most cancers, reducing the sequencing depth for miRNAs and mRNAs leads to similar performance, and the possible library size reduction is correlated between miRNAs and mRNAs when chosen as covariates in the Cox model (Appendix A). For mRNAs, one order of magnitude reduction or more is permitted for 11 cancers but CESC, which only tolerates a 50% reduction (∼20,000,000 aligned reads). For miRNAs, there are 3 exceptions: CESC again, which also tolerates a 50% reduction (∼2,000,000 aligned reads), PRAD with an 80% reduction (∼1,000,000 aligned reads), and LUAD which do not tolerate any sequencing depth reduction, and may even show improved performance with an increase in library size (≥5,000,000 aligned reads). For cancers tolerating 500,000 aligned reads in mRNAs or less, the sequencing depth could be reduced at least for one order of magnitude in miRNAs. Thus, mRNA sequencing data might inform of the required sequencing depth for miRNAs.

For random survival forest and mRNA-seq data, the sequencing depth of all cancers can be reduced by a factor 100 without degrading the C-index and the IBS (Appendix A). This fold reduction corresponds to median sequencing depth of about 500,000 aligned reads. For KIRC and LGG the sequencing depth can be even more reduced, by a factor 1000 (∼50,000 aligned reads). The results are more heterogeneous for miRNA-seq data as the sequencing depth can be reduced by a factor 1000 for CESC (∼5000 aligned reads) down to 5 for KIRP and MESO (∼1,000,000 aligned reads). Noticeably, for CESC, the possible fold reduction is much larger for random survival forest than for the Cox model for both miRNAs and mRNAs. We hypothesized that these differences are the consequence of a better C-index obtained with the Cox model than with random survival forest (Appendix A).

### 3.6. Models Trained with Fewer Patients Do Not Degrade Prognosis for Most of the Investigated Cancers

For the Cox model and miRNA-seq data, the number of patients in the training dataset can be reduced for 9 of the 11 cancers, at least for a small proportion (Appendix A). The two exceptions concern PRAD and LUAD, for which diminishing the number of patients in the training set decreases the prediction performance. We obtained comparable results for mRNA-seq data, except for UVM and UCEC which require more patients to achieve maximum performance—for UVM with similar C-index between miRNAs and mRNAs, but for UCEC with better performance with miRNAs (Appendix A). Surprisingly, random survival forest need less patients in the training dataset to achieve optimal prediction performance. This result makes it possible to consider stratifying patients into subgroups and to learn models separately for each subgroup.

### 3.7. Very Small Sequencing Depth Is Responsible for the Performance Loss

When reducing sequencing depth, we automatically reduce the number of detected genes. We define a non-coding/coding gene as ‘detected’ if its CPM-normalized expression level is greater than 1 for at least 1% of the patients in the training dataset. Two hypotheses can be put forward to explain the decrease in predictive abilities induced by the subsampling of sequencing depth:(1)the number of detected genes decreases as the subsampling rate increases (Appendix A, [38]), and only the level of expression of the most highly expressed genes can be measured (Appendix A). However, genes with a low level of expression may have significant predictive power and go undetected, which would diminish overall predictive capabilities.(2)more generally, the signal-to-noise ratio decreases for all genes (the standard deviation of the measurements varies in $\sqrt{N}$, with *N* the number of aligned reads per gene).

To test the first hypothesis, we compared the C-indices obtained with all miRNAs, and those obtained only with the most expressed genes, using the Cox model. This corresponds on average to a 2-fold decrease in the number of predictors (i.e., 210 miRNAs on average detected after subsampling of the miRNA-seq data by a factor of 10,000 for all cancers studied). In these two scenarios, the sequencing data keep the same read depth per gene, but only the number of genes taken into account differs. For miRNAs, reducing the dimension by keeping only the 210 most expressed genes does not significantly impact prediction capabilities (i.e., C-index and IBS) for the 11 cancers studied, except for CESC and LUAD with the C-index (*p*-value <0.001, one-sided Wilcoxon signed-rank test with Benjamini-Hochberg correction), and to a lesser extent for LIHC with the C-index (*p*-value <0.05, Appendix A). The first hypothesis is therefore not verified.

To test the second hypothesis, we calculated the C-indices obtained after subsampling by a factor of 10,000 (i.e., 210 genes on average are detected, and the count data are subsampled), and those obtained with the same 210 genes (on average) but without subsampling, also with the Cox model. For all 11 cancers, the median C-index obtained after subsampling is lower than that obtained without subsampling but with the same predictors (Appendix A). For the IBS, we observed the same results, except for UVM, LUAD and UCEC. The second hypothesis is verified: the subsampling induces a decrease in the signal-to-noise ratio of the RNA-seq count data explaining the decrease of the prediction if a strong subsampling is applied. It is also interesting to notice that for most cancers, selecting half of the most expressed predictors does not affect prediction performance.

We obtained comparable results for random survival forest (Appendix A) and for mRNA-seq data (data not shown to avoid too many supplementary figures, but can easily reproduced with the R scripts shared on github).

### 3.8. Prognostic Performances Follow Similar Trend after Subsampling When Tested on an Independent Dataset

To evaluate to which extent these results can be used in practice, we checked whether the models learned on TCGA, with an incremental decrease in sequencing depth, also do not degrade the predictions on an independent dataset. However, we were unable to find a dataset comparable to TCGA with both microRNA tumor profiling and patient survival, so we chose to demonstrate the reproducibility on mRNA profiling. We chose an mRNA profiling dataset measured with microarray in order to improve generalizability, and keeping ccRCC for the reasons detailed Section 3.5. Figure 2 shows that first the C-index is higher when the test set is the independent one, and second that the trend is comparable for both test sets (E-MTAB-1980 and TCGA datasets). More precisely, reducing the mRNA sequencing depth by a factor of 10 or even 100 on the training set does not affect the C-index performance. The E-MTAB-1980 dataset also provide improved IBS performance and smaller variability. We hypothesize that the smaller variability is due to the fact that the E-MTAB-1980 test set remains the same, as compared to the TCGA test set gathering the 20% of remaining patients, not used in the training set, and so the difference in performance is more sensitive (Appendix A). Halving the number *n* of patients in the training set (from 80% to 40%, i.e., from n=406 to 203) increases the IBS measured on the E-MTAB-1980 dataset in a modest manner (from a median of 0.112, 95% confidence interval of the median [0.11;0.114] to 0.133 [0.125;0.14]), whereas only 10% (n=51) of the patients lead to low IBS performance (0.205 [0.195;0.214]). This information is useful when trying to improve the model by clustering the patients into refined cancer subtypes, and further applying different Cox model in each of these subtypes: while 2 subtypes are reasonable for ccRCC on this large TCGA cohort, more than 5 may lead to too large IBS values only because of the small number of patients in each learning subset.

## 4. Discussion

Our work shows the benefit of using tumor mRNA or miRNA profiling to predict patient survival. We thus estimated the optimal sequencing depth and number of patients to achieve this prediction, using Cox and random forest models. We considered that the sequencing depth can be reduced if both C-index and IBS are not significantly degraded. This choice is subjective and can easily be adapted with the R script provided. For example, if discrimination among patients at risk is the only important aspect for a particular study, the C-index should be considered as the only prediction metrics. Second, as the same initial set of patients is used in multiple Monte-Carlo runs (10 repetitions of 5-fold cross-validation) to estimate prediction performance, the 50 metrics (i.e., C-index or IBS) are not independent. Besides, the *p*-values can be reduced toward 0 when small differences are observed by increasing the number of repetitions. The computed *p*-values are thus only indicative, but helps the readability of the graphics.

Thus, because of the methodology used, we obtain comparable results with slight differences with [13] about shallow tumor mRNA sequencing to predict patient survival with Cox model. We have extended it to miRNA-sequencing, and to random forest survival.

Then, estimation of the IBS with random survival forest is direct in the sense that the survival function *S* is an output of the algorithm for patients of the testing dataset. However, it is not the case with the Cox model as the baseline hazard function $h_{0}$ is not estimated in the pseudo-likelihood. This function $h_{0}$ is estimated with the Breslow estimator [14]. This dissimilarity in the way individual survival functions are estimated may explain the large advantage on prediction performance for random survival forest as compared to the Cox model (Appendix A). More investigations are still needed to better understand this observation.

C-index and IBS metrics do not always provide identical evaluation of the performance of the models. While the C-index only accounts for the ordering of the patients depending on the risk score, the IBS also evaluates the base line $h_{0}\left(t\right)$. We hypothesize that the quality of the inference of the baseline is responsible for the different behavior of both metrics. For specific goals, one of the metric could be favored. For example, to identify patients at risk, we encourage to focus on the C-index. In other cases, and as the two metrics are complementary, we advise to consider both metrics.

This work is the first one to focus on the sequencing depth of miRNA profiling for survival prediction, and could serve as a proxy to calibrate future experiments. Lung adenocarcinoma (LUAD) showed a noticeable difference with other cancers as tumor miRNAs better predict patient survival than mRNAs. This may indicate the particular role of miRNAs in this tumor type, and would be worth to further investigate.

Finally, the number of patients required in the training dataset is lower for random survival forest than for the Cox model (Section 3.6). This result is surprising as random forest has more degrees of freedom, and further work is needed to investigate this point.

## 5. Conclusions

In this work, we present a methodology and results on the possibility of (i) reducing sequencing costs to, for example, create validation datasets, and (ii) reducing the number of samples in training datasets to stratify patients into subgroups in the context of prediction of survival in cancer. Cox model and random survival forest provide comparable C-indices for some cancers but not all (e.g., the Cox model outperforms random survival forest for CESC and miRNA-seq data; the contrary being true for THCA). However, with IBS as the performance metric, the performance are better with random survival forest. We also pointed out that mRNA-seq data provide slightly better performance than miRNA-seq data on average, with the noticeable exception of lung adenocarcinoma (LUAD). Integration of miRNA-seq data with clinical data allows to improve predictions over clinical data alone for most of the 11 investigated cancers. Importantly, we demonstrated that sequencing depth of miRNA-seq and mRNA-seq can be reduced without degrading prediction performance for most of the 11 cancers retained in a cancer, data (i.e., miRNAs or mRNAs) and metric (i.e., C-index or IBS) dependent manner, thus allowing the reduction of sequencing cost to create independent validation datasets. Finally, we demonstrated that the number of patients in the training dataset can be reduced for both miRNA and mRNA data without degrading the prediction performance for the Cox model, and in a larger extent for random survival forest. Finally, our results were confirmed on an independent dataset.

## Figures and Tables

**Figure 1 genes-13-02275-f001:**
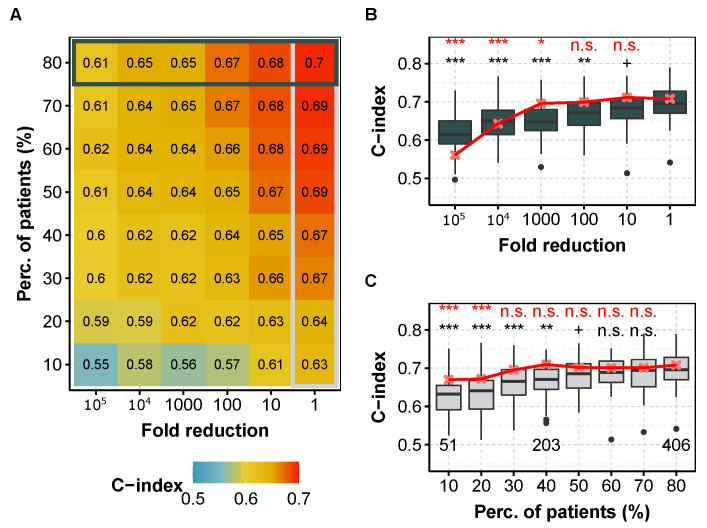
C-index obtained for different fold reduction factors and percentage of patients in the training dataset for KIRC (ccRCC, TCGA) with the Cox model. (**A**) Median C-index for different degradation of both sequencing depth (x axis) and percentage of patients (y axis) in the training dataset for miRNA-seq data. Horizontal box highlights the case where all 80% of patients are used and corresponds to (**B**), whereas vertical box focuses on the full available library size and corresponds to (**C**). (**B**) C-index for different fold reduction factors for miRNA-seq (gray boxplots) and mRNA-seq data (median values, in red) with 80% of the patients in the training dataset. Above is the *p*-value of a one-sided Wilcoxon test compared to no subsampling (i.e., δ=1). (**C**) C-index for different percentages of patients in the training dataset for miRNA-seq (light gray boxplots) and mRNA-seq data (median values, in red) with original TCGA sequencing depth. Above is the *p*-value compared to the full dataset (i.e., 80%). red, mRNA-seq; gray boxplots, miRNA-seq. In each case, we computed the C-indices by 10 repetitions of 5-fold cross validation. ***: p≤0.001, **: p≤0.01, *: p≤0.05, +: p<0.1, n.s.: p≥0.1.

**Figure 2 genes-13-02275-f002:**
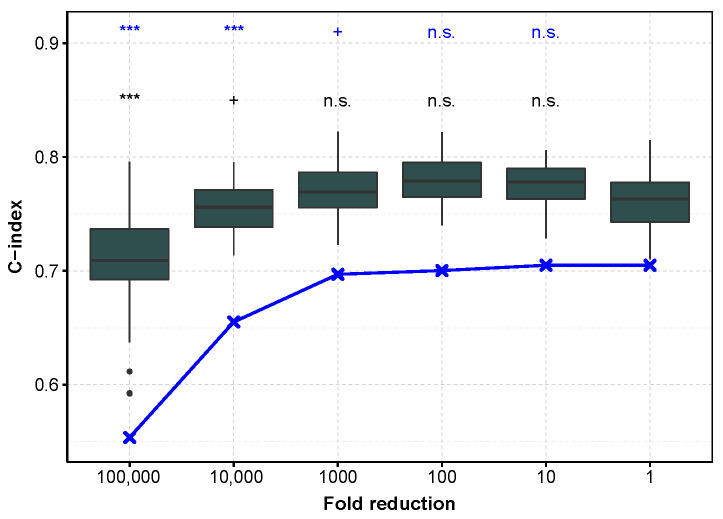
C-index as a function of sequencing depth reduction tested on the E-MTAB-1980 dataset and TCGA subset for mRNA profiling in ccRCC. In dark gray, performance measured with the C-index calculated on the E-MTAB-1980 dataset, after training on an 80% sub-sample of the TCGA dataset (the procedure is repeated to obtain 50 C-indices). In blue, the test is performed on the remaining 20% of TCGA data (median C-index). ***: p≤0.001, +: p<0.1, n.s.: p≥0.1.

**Table 1 genes-13-02275-t001:** Characteristics of the 11 cancers investigated. We computed the C-indices with 10 repetitions of 5-fold cross-validation for both the Cox-elastic net model (EN) and random survival forest (RF). Datasets are ordered according to their median C-index computed with Cox-elastic net model (decreasing order).

Cancer	n Patients	p miRNA	Censoring Rate	Survival— 3 Years	C-Index— EN	C-Index— RF
UVM	77	536	0.73	0.74	0.81	0.83
ACC	77	518	0.65	0.75	0.8	0.84
KIRP	269	486	0.84	0.87	0.79	0.82
MESO	85	519	0.14	0.19	0.7	0.69
KIRC	508	462	0.66	0.75	0.7	0.66
LGG	506	548	0.62	0.56	0.7	0.69
CESC	288	542	0.76	0.72	0.68	0.59
LIHC	355	540	0.65	0.62	0.67	0.66
PRAD	486	470	0.81	0.8	0.66	0.59
LUAD	483	529	0.63	0.61	0.66	0.6
UCEC	532	554	0.83	0.83	0.61	0.64

**Table 2 genes-13-02275-t002:** Maximum miRNA-seq library size reduction before the decreasing of prediction performance, corresponding median sequencing depth (in thousands of aligned reads), and prediction metric degraded first, for the Cox model, and the 11 investigated cancers.

Cancer	UVM	ACC	KIRP	MESO	KIRC	LGG	CESC	LIHC	PRAD	LUAD	UCEC
Fold reduction	1000	1000	100	100	10	10	2	10	5	< 1	10,000
Median library size (in 1000 reads)	5	6	60	50	200	700	2000	500	900	> 5000	1
Metric degraded first	C-index	both	both	C-index	both	IBS	C-index	both	C-index	both	C-index

**Table 3 genes-13-02275-t003:** Maximum mRNA-seq library size reduction before the decreasing of prediction performance, corresponding median sequencing depth (in thousands of aligned reads), and prediction metric degraded first, for the Cox model, and the 11 investigated cancers.

Cancer	UVM	ACC	KIRP	MESO	KIRC	LGG	CESC	LIHC	PRAD	LUAD	UCEC
Fold reduction	100	1000	100	100	10	100	2	10	10	10	10
Median library size (in 1000 reads)	400	40	400	500	5000	500	20,000	5000	5000	4000	2000
Metric degraded first	C-index	both	IBS	both	IBS	both	C-index	IBS	both	IBS	both

## Data Availability

Publicly available datasets were analyzed in this study. These data can be found here: https://gdac.broadinstitute.org/ https://www.ebi.ac.uk/arrayexpress/experiments/E-MTAB-1980/. The R script to reproduce the results presented in this article is available at https://github.com/remyJardillier/Survival_seq_depth (all accessed on 24 October 2022).

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
