# Peer review of "Optimal microRNA Sequencing Depth to Predict Cancer Patient Survival with Random Forest and Cox Models"

_genes, 2022, doi:10.3390/genes13122275_

Round 1

Reviewer 1 Report

This paper discovers optimal microRNA sequencing depth to predict cancer patient survival with random forest and Cox models. The method is proposed with certain novelty, and utilized with adequate soundness. However, it will be better if the authors can draw a figure or flowchart to illustrate the proposed method.

Reviewer 2 Report

This paper investigates the impact of sequencing depth and training sample sizes of miRNA and mRNA on the predictive power of cancer survival. It is a comprehensive study that includes 25 cancer types, 11 of which are involved in all aspects of the investigation. The authors applied k-fold cross-validation on each cancer type. They evaluated the survival predictivity in terms of C-index and IBS scores of two models, the Cox model and the random survival forest. As a primary result, they found that decreasing the sequencing depths or the number of training samples at a certain degree (magnitude) did not hurt performances on most cancer types. Additionally, they evaluated an external E-MTAB-1980 dataset and claimed that trained models with down-sampled reads and training samples in TCGA wouldn't lead to inaccurate survival predictions on the ccRCC E-MTAB-1980 dataset.  

It is an interesting study and would supply valuable insights to researchers to reduce the costs of performing miRNA and mRNA sequencing of cancer subtypes while not losing important biological information. However, the authors must address the following concerns to make the work more rigorous for publication. 

Major comments: 

  1. 1. It is unclear, at least from the main text (Section 2.5.1), whether the reads subsampling was only applied to the training data or both training and testing data. This is important because the evaluation of the original unsampled data is not meaningful at all in this case. Any conclusions drawn from the otherwise are not valid. Please clarify this point. 

  2.  
  1. 2. As sequencing depths and the number of patients are the two major factors investigated in the paper, the following points need to be clarified: 

  1. a. 11 out of 25 cancers were chosen for all aspects of the study in the paper based on whether at least one algorithm showed a median C-index significantly higher than 0.6 (lines 132-136). The author needs to clarify if the sequence-depths/#patients statistically differ between the selected and dropped-out cancers. 

  1. b. For the external E-MTAB dataset, it is not clear what the sequencing depth and the number of patients are. It would be great if the authors showed the results based on the test data with the same magnitudes of sequencing depths of the training data at each subsampling level. 

  1. 3. As C-index and IBS did not agree entirely in most of the experiments, the authors might briefly discuss the cause of the discordance and which is a better metric based on their experience and the results. Such a discussion would give other researchers a better understanding of which metric to use. 

Minor comments: 

  1. 1. Line 32-34, it would be great if the author could discuss the differences among different regularization functions with references, e.g., Lasso, Ridge, and elastic, and present an argument of why the elastic penalty was chosen based on the terms' characteristics. For example, L1 might overly reduce the number of features.  

  1. 2. Line 339 explains why mRNA results (data) are not shown.  

  1. 3. Line 223, 'vas' was a typo. It should be 'was.'

Round 2

Reviewer 2 Report

The authors have addressed all my concerns and comments.